# Aspirin or enoxaparin for VTE prophylaxis after primary partial, total or revision hip or knee arthroplasty: A secondary analysis from the CRISTAL cluster randomized trial

**The CRISTAL Study Group**[¶]*

¶ Members of author group included in Acknowledgments.
* verinder.s.sidhu@gmail.com

**Data Availability Statement:** Data cannot be shared publicly because of legislative restrictions on the use of registry data from the Australian

## Abstract

### Background

This study compares aspirin to enoxaparin for symptomatic VTE prophylaxis within 90 days of any type of hip or knee arthroplasty performed for any diagnosis, in patients enrolled in the CRISTAL trial.

### Materials and methods

CRISTAL was a cluster-randomised crossover, registry-nested non-inferiority trial across 31 hospitals in Australia. The primary publication was restricted to patients undergoing primary total hip or knee arthroplasty for a diagnosis of osteoarthritis. This report includes all enrolled patients undergoing hip or knee arthroplasty procedures (partial or total, primary or revision) performed for any indication. Hospitals were randomized to administer patients aspirin (100mg daily) or enoxaparin (40mg daily), for 35 days after hip arthroplasty and 14 days after knee arthroplasty. Crossover occurred after the patient enrolment target had been met for the first group. The primary outcome was symptomatic VTE within 90 days. Analyses were performed by randomization group.

### Results

Between April 20, 2019 and December 18, 2020, 12384 patients were enrolled (7238 aspirin group and 5146 enoxaparin). Of these, 6901 (95.3%) given aspirin and 4827 (93.8%) given enoxaparin (total 11728, 94.7%) were included in the final analyses. Within 90 days, symptomatic VTE occurred in 226 (3.27%) aspirin patients and 85 (1.76%) enoxaparin patients, significant for the superiority of enoxaparin (estimated treatment difference 1.85%, 95% CI 0.59% to 3.10%, p = 0.004). Joint-related reoperation within 90 days was lower in the enoxaparin group (109/4827 (2.26%) vs 171/6896 (2.47%) with aspirin, estimated difference 0.77%; 95% CI 0.06% to 1.47%, p = 0.03). There were no significant differences in the other secondary outcomes.

Orthopaedic Association National Joint Replacement Registry (AOANJRR). Data are available from the AOANJRR for those who meet the criteria for access. External access to and use of de-identified AOANJRR data and CRISTAL data is permitted but must be in accordance with AOANJRR policies (Ref No POL.S3.3, S3.4, S3.5) available on the registry website: https://aoanjrr.sahmri.com/ policies. Requests for data use can be made by contacting the AOANJRR: Telephone: +618 8128 4284 Email: adhocs@aoanjrr.org.au.

**Funding:** The trial was funded by an MRFF Grant provided by the Australian Federal Government (grant number 1152285)

**Competing interests:** The authors have declared that no competing interests exist

## Conclusion

In patients undergoing hip or knee arthroplasty (of any type, performed for any indication) enrolled in the CRISTAL trial, aspirin compared to enoxaparin resulted in a significantly higher rate of symptomatic VTE and joint-related reoperation within 90 days. These findings extend the applicability of the CRISTAL trial results.

## Trial registration

Anzctr.org.au, identifier: ACTRN12618001879257.

## Introduction

Venous thromboembolism (VTE) remains a serious complication following hip and knee arthroplasty and occurs in approximately 2% of patients [1–3]. The use of low-dose aspirin monotherapy as chemoprophylaxis has increased throughout Australia and the United States since 2010, despite a lack of evidence from randomised trials [4, 5].

The primary results of the CRISTAL study demonstrated that enoxaparin was superior to aspirin in the prevention of symptomatic VTE within 90 days of primary total hip and knee arthroplasty performed for a diagnosis of osteoarthritis, in patients eligible to receive the study drug [3]. This report expands the scope of the primary study by reporting on a prespecified analysis [6] and including all patients enrolled into CRISTAL undergoing any hip or knee arthroplasty procedure (partial, total, primary or revision) for any diagnosis.

## Materials and methods

### Trial design

The design, protocol and statistical analysis plan of CRISTAL has previously been published [3, 6, 7]. CRISTAL was a cluster-randomised, crossover, non-inferiority trial performed across 31 hospitals in Australia, nested within the Australian Orthopaedic Association National Joint Replacement Registry (AOANJRR). The trial was approved by the Sydney Local Health District human research ethics committee (lead, number X18-0424, date 11th of December, 2018) and by all ethic committees prior to commencement. It was funded by a Medical Research Futures Fund Grant from the Australian Federal Government, who had no role in the design of the trial or the analysis and interpretation of the results.

All enrolled patients provided consent for provision and use of their data, but a consent waiver was granted by each ethics committee for the use of either study drug, as both were commonly used for prophylaxis throughout Australia prior to trial commencement. The trial was monitored by a trial management committee

### Trial participants

The clusters were defined as the participating hospitals. Hospitals were eligible if they had performed >250 hip or knee arthroplasty procedures in the year prior to recruitment and if they agreed to follow the trial protocol. All adult patients (aged ≥ 18 years) undergoing hip or knee arthroplasty procedures were eligible for inclusion. Patients not eligible to receive either study drug were those on long-term preoperative anticoagulants (specifically a direct oral anticoagulant (DOAC), warfarin or dual-antiplatelet therapy (DAPT)) or those with a medical

contraindication (allergy or bleeding disorder precluding anticoagulation). These patients were still included in the analyses, but did not receive either study drug for VTE prophylaxis. Patients on preoperative long-term single antiplatelet medication received the study drug as allocated.

The primary paper was restricted to patients undergoing primary total hip or knee arthroplasty for osteoarthritis. This paper reports a prespecified analysis [6] and includes all enrolled patients undergoing any arthroplasty procedure performed for any indication.

### Randomization and blinding

Hospitals were allocated to consecutive periods of a standard protocol of enoxaparin or aspirin, with the initial treatment order being randomized. Hospitals were randomized in permuted blocks of size four by statisticians from the South Australian Health and Medical Research Institute, independent of study investigators. Hospitals were advised to crossover once the sample size for the first treatment arm was met, which was monitored by AOANJRR staff.

Participating hospitals were not blinded to treatment allocation. Patients were aware that they were participating in a trial comparing different treatments for VTE prophylaxis, but were unaware of specific study details and whether they were in the intervention or control group. Study investigators and the DSMB were blinded to treatment assignment during the trial and all analyses.

### Interventions and assessment

Patients were consented and enrolled at pre-admission clinic appointments by nursing staff or treating surgeons. Patients in the aspirin group received aspirin at 100mg once daily, orally for 35 days post hip arthroplasty and for 14 days post knee arthroplasty, commencing within 24 hours postoperatively. Patients in the enoxaparin group received enoxaparin at 40mg daily, subcutaneously for the same time periods, reduced to 20mg for patients weighing less than 50kg and for patients with an estimated glomerular filtration rate <30mL/min. Other standard interventions across sites were intra and postoperative intermittent pneumatic compression calf devices until patients were mobile, compression stockings, and mobilisation offered on day 0 or 1 postoperatively.

Data was collected for each cluster and each enrolled patient and methods of data collection have been published previously [6, 7]. All patients who responded 'yes' to having experienced a VTE, a secondary operation within 90 days or 6 months or a major bleeding event within 90 days had this outcome verified from treating doctors.

### Outcome measures

The primary outcome of the study was symptomatic VTE within 90 days of surgery. Screening tests for VTE were not performed in asymptomatic patients. Secondary outcomes were joint-related reoperation, joint-related readmission, major bleeding events (defined as those resulting in readmission, reoperation or death) and all-cause mortality within 90 days and joint-related reoperation within 6 months.

### Interim analysis

An interim analysis was not initially planned. Due to a serious adverse event at one site, a Data Safety Monitoring Board (DSMB) was convened and an interim analysis conducted. The trial management committee applied the Haybittle-Peto stopping rule of a two-sided significance of 0.001, to detect a between-group superiority difference for the primary outcome [26, 27].

After the first interim analysis, the DSMB recommended trial continuation and a second analysis three months later. The second interim analysis demonstrated that the trial stopping rule had been met and patient recruitment was ceased at all sites upon advice of the DSMB.

### Statistical analyses

All analyses were performed using SAS version 9.4 (SAS Institute, Cary USA) and R (R Foundation for Statistical Computing Platform) version 4.1.0 and used an intention-to-treat principle.

The sample size calculation was performed for the primary study (restricted to patients undergoing total hip or knee arthroplasty for a diagnosis of osteoarthritis, eligible to receive the study drug). This used a non-inferiority margin of 1% (based on the current available literature) [4, 8], a power of 90% and a one-sided significance level of 0.025. For a cluster-randomised crossover trial with an intracluster correlation of 0.01, an interperiod correlation of 0.008 and 31 clusters, this yielded 11,160 patients (180 patients per group for each cluster). This current report included a broader population than the primary analysis and was assumed to be powered sufficiently to determine whether aspirin was non-inferior to enoxaparin for all included patients.

The analysis for the primary outcome tested the between-group difference of patients developing a symptomatic VTE within 90 days for non-inferiority at a margin of 1%. Between-group differences were estimated for the primary and secondary outcomes using cluster summary methods, accounting for unequal cluster sizes and incomplete crossover of some clusters following the early stopping of the trial. For the primary outcome, secondary outcomes and secondary analyses, 95% confidence intervals were examined to determine whether superiority could be concluded.

Multiple imputation for missing data from loss to follow-up used the R package 'jomo' to account for clustering through multilevel joint modelling. Date of death was linked from the National Death Index and was complete.

An analysis was conducted for the primary outcome in the subpopulation of patients who were eligible to receive the study drug (those on a DOAC, warfarin or DAPT or those with a medical contraindication as above), undergoing any hip or knee arthroplasty procedure.

A post-hoc exploratory analysis was performed to compare the cause of joint-related reoperation within 90 days between groups. The causes of joint-related reoperation were divided into six categories: infection, manipulation under anaesthesia (MUA), wound complications, dislocation, fracture or other. This was performed for all enrolled patients and for all enrolled patients eligible to receive the study drug.

### Results

Thirty-one hospitals were recruited between April 15th, 2019 and August 12th, 2019. Sixteen crossed over prior to trial cessation (11 crossed over to aspirin and 5 to enoxaparin), and the remaining 15 hospitals did not cross over. Prior to trial cessation, 13717 patients were enrolled between April 20th, 2019 and December 18th, 2020, with 12384 patients eligible for inclusion in this study (Fig 1).

There were 11728 patients (94.7%) patients included with complete data for the primary outcome, 6901 (95.3%) in the aspirin group (from 27 hospitals) and 4827 (93.8%) in the enoxaparin group (from 20 hospitals). Exclusions from the final analysis are shown in Fig 1 but this did not differ by group.

The two groups were similar on baseline variables (Table 1).

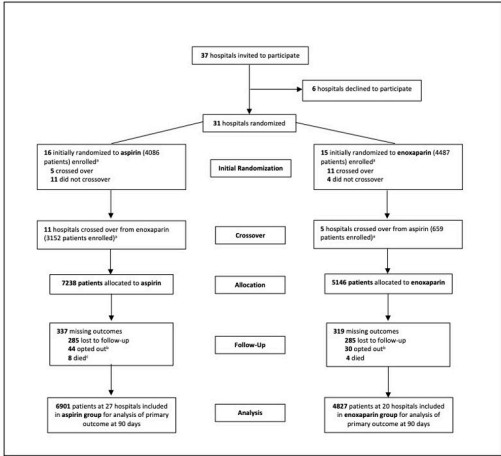

**Fig 1. CONSORT flow diagram of participating hospitals and patients.** [a]A total of 1333 patients declined to have their data collected after enrolment and therefore group allocation was not recorded. These patients were not included in any analysis and are not included in the tallies represented here. [b]Patients who opted out chose to withdraw after initially consenting and prior to 90-day follow-up. [c]One additional patient who died recorded a venous thromboembolic event prior to death and therefore was not considered to be missing an outcome.

## Primary outcome

Imputed results are reported. Symptomatic VTE within 90 days occurred in 226 of 6901 patients (3.27%) in the aspirin group and in 85 of 4827 patients (1.76%) in the enoxaparin group (estimated treatment difference, 1.85%, 95% confidence interval [CI], 0.59% to 3.10%) (Table 2). This was significant for the superiority of enoxaparin (p = 0.004).

## Secondary outcomes

Death within 90 days occurred in 9 of 7238 patients (0.12%) in the aspirin group and in 4 of 5146 patients (0.08%) in the enoxaparin group (estimated difference, 0.06%, 95% CI, -0.05% to 0.17%, p = 0.28). Major bleeding events occurred in 20 of 6880 (0.29%) patients in the aspirin group and 25 of 4818 patients (0.52%) in the enoxaparin group (estimated difference -0.18%; 95% CI, -0.52% to 0.17%, p = 0.31). Joint-related reoperation within 90 days occurred in 171 of 6896 patients (2.47%) in the aspirin group and in 109 of 4827 patients (2.26%) in the enoxaparin group (estimated difference 0.77%; 95% CI 0.06% to 1.46%, p = 0.03). There was no significant between-group difference for joint-related re-admission within 90 days or joint-related reoperation within 6 months (Table 2).

## Subpopulation analysis

There were 10960 patients enrolled who were eligible to receive the study drug and 10381 completed 90 day follow-up (94.7%). Symptomatic VTE within 90 days occurred in 202 of 6087 patients (3.32%) in the aspirin group and in 73 of 4294 patients (1.70%) in the enoxaparin group (estimated difference 1.85%, 95% CI 0.60% to 3.31%, p = 0.005).

## Reasons for joint-related reoperation within 90 days

A comparison of reasons for joint-related reoperation within 90 days for all enrolled patients and for all enrolled patients eligible to receive the study drug by assigned group is shown in

**Table 1. Baseline patient characteristics for the study population, according to treatment allocation.**

| | Aspirin (n = 7238) | Enoxaparin (n = 5146) |
|---|---|---|
| Median age (IQR), years | 68.0 (61.0–74.0) | 69.0 (61.0–75.0) |
| Median body-mass index (IQR), kg/m$^2$ | 30.5 (26.8–35.1) | 30.4 (26.8–34.8) |
| | (n = 7151) | (n = 5079) |
| Female sex, n (%) | 4008 (55.6) | 2859 (55.6) |
| American Society of Anesthesiologists Classification, n (%) | n = 7238 | n = 5146 |
| 1 | 369 (5.1) | 242 (4.7) |
| 2 | 3820 (53.0) | 2658 (51.7) |
| 3 | 2932 (40.7) | 2192 (42.6) |
| 4 | 90 (1.2) | 51 (1.0) |
| Previous venous thromboembolism, n (%) | 545/6812 (8.0) | 404/4897 (8.2) |
| Long term single antiplatelet therapy, n (%) | n = 6285 | n = 4510 |
| Aspirin | 961 (15.3) | 706 (15.7) |
| Other single antiplatelet | 110 (1.8) | 68 (1.5) |
| Other agent (unspecified) | 252 (4.0) | 173 (3.8) |
| Ineligible to Receive Study Drug, n (%) | | |
| Direct oral anticoagulant | 408 (5.6) | 285 (5.5) |
| Warfarin | 88 (1.2) | 48 (0.9) |
| Dual antiplatelet therapy | 26 (0.4) | 28 (0.5) |
| Medical contraindication | 327 (4.5) | 182 (3.5) |
| Joint Replacement, n (%) | | |
| Hip Arthroplasty | 2790 (38.5) | 2045 (39.7) |
| Knee Arthroplasty | 4448 (61.5) | 3101 (60.3) |
| Bilateral simultaneous arthroplasty, n (%) | 760 (10.5) | 507 (9.9) |
| Type of Surgery, n (%) | | |
| Primary Total | 6722 (92.9) | 4762 (92.5) |
| Primary Partial | 188 (2.6) | 133 (2.6) |
| Primary Resurfacing | 30 (0.4) | 31 (0.6) |
| Revision | 298 (4.1) | 220 (4.3) |
| Indication, n (%) | | |
| Osteoarthritis | 6644 (91.8) | 4674 (90.8) |
| Avascular Necrosis | 119 (1.6) | 97 (1.9) |
| Inflammatory | 83 (1.1) | 51 (1.0) |
| Fracture | 22 (0.3) | 52 (1.0) |
| Other | 370 (5.2) | 275 (5.3) |
| Prosthesis, n (%) | n = 7229 | n = 5141 |
| Cemented | 3536 (48.9) | 2607 (50.7) |
| Hybrid | 1790 (24.8) | 1357 (26.4) |
| Uncemented | 1903 (26.3) | 1177 (22.9) |

Table 3. Reasons for joint-related reoperation did not differ between study groups irrespective of eligibility to receive the study drug.

## Discussion

Enoxaparin was found to be superior to aspirin for symptomatic VTE prophylaxis within 90 days of hip or knee arthroplasty (including partial, total or revision procedures) performed for any indication. Enoxaparin was associated with a lower joint-related reoperation rate within

**Table 2. Primary and secondary outcomes via treatment allocation.**

| | Number/total (%) | | Estimated Absolute Risk Difference (%) | 95% Confidence Interval (%) | p-value[a] |
|---|---|---|---|---|---|
| | **Aspirin** | **Enoxaparin** | | | |
| | **(n = 6901)** | **(n = 4827)** | | | |
| **Primary outcome** | | | | | |
| **Any symptomatic venous thromboembolism within 90d** | 226/6901 (3.27) | 85/4827 (1.76) | 1.85 | 0.59 to 3.10 | 0.004 |
| **Components of primary outcome** | | | | | |
| **Pulmonary embolism within 90d** | 69/6901 (1.00) | 27/4827 (0.56) | 0.40 | -0.18 to 0.98 | 0.18 |
| **Any deep venous thrombosis within 90d** | 169/6901 (2.45) | 60/4827 (1.24) | 1.51 | 0.52 to 2.50 | 0.003 |
| **Above knee deep venous thrombosis within 90d** | 14/6901 (0.20) | 10/4827 (0.21) | -0.05 | -0.23 to 0.13 | 0.60 |
| **Both pulmonary embolism and any deep venous thrombosis within 90d** | 12/6901 (0.17) | 2/4827 (0.04) | 0.07 | (-0.19 to 0.33) | 0.59 |
| **Secondary outcomes** | | | | | |
| **Death within 90d** | 9/7238 (0.12) | 4/5146 (0.08) | 0.06 | -0.05 to 0.17 | 0.28 |
| **Major bleeding within 90d** | 20/6880[b] (0.29) | 25/4818 (0.52) | -0.18 | -0.52 to 0.17 | 0.31 |
| **Re-admission within 90d** | 196/6883 (2.85) | 133/4823 (2.76) | 0.52 | -0.18 to 1.22 | 0.14 |
| **Re-operation within 90d** | 171/6896 (2.48) | 109/4827 (2.26) | 0.77 | 0.06 to 1.47 | 0.03 |
| **Re-operation within 6 months** | 275/6460 (4.26) | 183/4522 (4.05) | 0.72 | -0.26 to 1.70 | 0.15 |

[a] p-values listed are for superiority

[b] Denominators differ by category due to missing data for secondary outcomes

90 days, compared to aspirin, and did not result in higher rates of major bleeding, readmission or mortality within 90 days. These results are comparable to those reported in the primary analysis.

There have been two retrospective studies comparing aspirin to other forms of prophylaxis for symptomatic VTE prevention following hip or knee arthroplasty since the primary CRISTAL publication, with no additional randomized trials published. The first study compared aspirin to DOACs, LMWH or warfarin and reported that aspirin was non-inferior to LMWH for the prevention of VTE within 90 days following hip or knee arthroplasty procedures [8]. Neither the type of LMWH used nor the dose was reported. The second study compared rivaroxaban to aspirin or LMWH, reporting that aspirin was superior to rivaroxaban in the prevention of pulmonary embolism and deep venous thrombosis [9]. Both studies attempted to adjust for confounders through analyses, however, the allocation of each agent was at the discretion of the treating physician and important data for each prophylaxis group (such as loss to follow-up, medication adherence and time to mobilization) were not available.

While there have been no further randomized trials published investigating aspirin following hip or knee arthroplasty, one published study has compared its use to enoxaparin for VTE prophylaxis following orthopaedic trauma. This trial reported similar results to those found in the primary CRISTAL publication [3] and those found in this study: symptomatic VTE rates were higher in patients allocated to aspirin, but there was no between-group difference observed in other outcomes, including mortality, within 90 days [10].

**Table 3. Reasons for joint-related reoperation within 90 days.**

| | Number (%) | | Estimated Absolute Risk Difference (%) | 95% Confidence Interval (%) | p-value[a] |
|---|---|---|---|---|---|
| | **Aspirin** | **Enoxaparin** | | | |
| | **(n = 6901)** | **(n = 4827)** | | | |
| **For All Enrolled Patients** | | | | | |
| **Infection** | 55/6896 (0.79) | 21/4827 (0.44) | 0.33 | -0.12 to 0.77 | 0.15 |
| **Manipulation Under Anaesthesia** | 39/6896 (0.57) | 41/4827 (0.85) | -0.02 | -0.50 to 0.46 | 0.94 |
| **Wound Complications** | 16/6896 (0.23) | 10/4827 (0.21) | 0.02 | -0.29 to 0.33 | 0.90 |
| **Dislocation** | 10/6896 (0.15) | 10/4827 (0.21) | -0.07 | -0.37 to 0.22 | 0.62 |
| **Fracture** | 17/6896 (0.25) | 10/4827 (0.21) | 0.10 | -0.24 to 0.43 | 0.57 |
| **Other** | 46/6896 (0.67) | 22/4827 (0.46) | 0.39 | 0.02 to 0.77 | 0.04 |
| **For all Enrolled Patients Eligible to Receive the Study Drug[b]** | **Aspirin** | **Enoxaparin** | | | |
| | **(n = 6901)** | **(n = 4827)** | | | |
| **Infection** | 49/6084 (0.81) | 19/4294 (0.44) | 0.27 | -0.15 to 0.70 | 0.21 |
| **Manipulation Under Anaesthesia** | 33/6084 (0.54) | 37/4294 (0.86) | -0.02 | -0.62 to 0.58 | 0.96 |
| **Wound Complications** | 16/6084 (0.26) | 9/4294 (0.21) | 0.13 | -0.19 to 0.45 | 0.42 |
| **Dislocation** | 10/6084 (0.16) | 9/4294 (0.21) | -0.05 | -0.37 to 0.27 | 0.76 |
| **Fracture** | 13/6084 (0.21) | 8/4294 (0.19) | 0.13 | -0.16 to 0.42 | 0.37 |
| **Other** | 39/6084 (0.64) | 20/4294 (0.47) | 0.23 | -0.16 to 0.63 | 0.25 |

[a]p-values listed are for superiority

[b]Patients eligible to receive the study drug were those not already on preoperative anticoagulation (specifically warfarin, direct oral anticoagulants or dual antiplatelet therapy) or those without a contraindication (e.g. allergy or medical contraindication)

A number of retrospective studies have also reported that aspirin results in lower rates of wound drainage or haematoma, stiffness, manipulation under anaesthesia (MUA) and deep prosthetic infection compared to other forms of prophylaxis due to its perceived lower risk of bleeding and wound drainage [11–13]. This was not supported by the results of this study, which found a lower joint-related reoperation rate for enoxaparin within 90 days. The primary analysis [7] and the recent randomised trial in trauma patients [10] also did not report lower re-operation rates with the use of aspirin compared to enoxaparin.

There have now over 25 retrospective studies [8, 9, 14–37] investigating aspirin monotherapy for symptomatic VTE prophylaxis following hip or knee arthroplasty since 2005 and only one published randomized trial over the same time period [3]. Retrospective studies are prone to bias as they provide poor control over the exposure of interest, covariates and potential confounders [38]. The majority of retrospective studies have included patients over a long time period, with some including patients from as early as 2000 in their cohorts [14]. Despite this, no retrospective study has conducted a temporal analysis to evaluate time as a potential confounder. Multiple studies have demonstrated that complications following hip and knee arthroplasty have decreased over the course of the last 20 years [39–41]. During this time,

aspirin use for VTE prophylaxis (compared to other agents) has increased [4, 5]. It is possible that the lower VTE risk and lower risk of joint-related reoperation in these observational studies may be attributed to general improvements in perioperative care over time rather than aspirin use.

This study broadens the scope and increases generalizability of the primary CRISTAL analysis, which was restricted to primary total hip or knee arthroplasty for a diagnosis of osteoarthritis only. Its larger sample size increases the power, precision and applicability of the results reported in the primary study. Despite the greater incidence of pulmonary embolism in the aspirin group found in the present analysis, we recognise that the study was likely underpowered for this outcome and for the outcome of proximal DVT. We also recognise that although the secondary outcome of joint-related reoperation included all relevant procedures (MUA, wound complications, fracture, deep prosthetic infection, prolonged drainage), an analysis by type of reoperation was not performed. Further, this result should be interpreted with caution since five secondary outcomes were analysed without a multiplicity adjustment and the result may be due to chance.

## Conclusions

Aspirin compared to enoxaparin resulted in a significantly higher rate of symptomatic VTE within 90 days in patients undergoing hip or knee arthroplasty of any type performed for any diagnosis. Enoxaparin did not result in a higher rate of secondary complications (major bleeding, death or readmission) and was found to be associated with a lower joint-related reoperation within 90 days. These findings increase the generalizability of the primary study to the full breadth of hip and knee arthroplasty procedures.

## Supporting information

**S1 Table. CONSORT 2010 checklist of information to include when reporting a cluster randomised trial.**
(DOCX)

**S1 File.**
(PDF)

## Acknowledgments

**Full List of Authors and Affiliations (all authors met the criteria for authorship as listed by the ICMJE):**

Verinder Sidhu 1,2; Thu-Lan Kelly 3; Nicole Pratt 3; Stephen E. Graves 4; Rachelle Buchbinder 5; Sam Adie 6; Kara Cashman 7; Ilana N Ackerman 5; Durga Bastiras 4; Roger Brighton 8, 9; Alexander W R Burns 10; Beng Hock Chong 11, 12; Helen Jentz 13; Maggie Cripps, 1, 2; Mark Dekkers 14; Richard de Steiger 15; Michael Dixon 16; Andrew Ellis 17, 18; Elizabeth C Griffith 7; David Hale 19; Amber Hansen 1, 2; Anthony Harris 20; Raphael Hau 15, 21; Mark Horsley 22; Dugal James 23; Omar Khorshid 24; Leonard Kuo 25; Peter Lewis 26; David Lieu 27; Michelle Lorimer 7; Samuel Macdessi 6, 28; Peter McCombe 29; Catherine McDougall 30; Jonathan Mulford 31; Justine Maree Naylor 1, 2; Richard S Page 32; John Radovanovic 33; Michael Solomon 34; Rami Sorial 35; Peter Summersell 36; Phong Tran 37; William L. Walter 17, 18, 38; Steve Webb 5; Chris Wilson 39, 40; David Wysocki 41; Ian A. Harris 1, 2, 42.

**Affiliations of Authors**

1. School of Clinical Medicine, UNSW Medicine & Health, South West Sydney Clinical School, Faculty of Medicine and Health, UNSW Sydney, Sydney, NSW, Australia

2. Whitlam Orthopaedic Research Centre, Ingham Institute for Applied Medical Research

3. Clinical and Health Sciences, Quality Use of Medicines Pharmacy Research Centre, University of South Australia, Adelaide, South Australia, Australia

4. Australian Orthopaedic Association National Joint Replacement Registry, Adelaide, South Australia, Australia

5. School of Public Health and Preventive Medicine, Monash University, Melbourne, Victoria, Australia

6. School of Clinical Medicine, UNSW Medicine & Health, St George & Sutherland Clinical Campuses, Faculty of Medicine and Health, UNSW Sydney, NSW, Australia

7. South Australian Health and Medical Research Institute, Adelaide, South Australia, Australia

8. Orthopaedic Department, Westmead Private Hospital, Westmead, Sydney, NSW, Australia

9. Orthopaedic Department, Lakeview Private Hospital, Baulkham Hills, Sydney, NSW, Australia

10. Orthopaedic Department, Calvary John James Hospital, Deakin, Canberra, ACT, NSW, Australia

11. Department of Medicine, Faculty of Medicine, University of New South Wales, Sydney, NSW, Australia

12. Department of Hematology, New South Wales Pathology, Kogarah Campus, Sydney, NSW, Australia

13. Musculoskeletal Australia, Melbourne, Victoria, Australia

14. Orthopaedic Department, Greenslopes Private Hospital, Greenslopes, Brisbane, QLD, Australia

15. Department of Surgery, Epworth Healthcare, University of Melbourne, Melbourne, Victoria, Australia

16. Orthopaedic Department, Kareena Private Hospital, Sutherland, Sydney, NSW, Australia

17. Orthopaedic Department, Royal North Shore Hospital, St Leonard's, Sydney, NSW, Australia

18. Sydney Musculoskeletal Health Flagship Centre of the University of Sydney and Royal North Shore Hospital, St Leonard's, Sydney, NSW, Australia

19. Orthopaedic Department, Hornsby and Kuringai Hospital, Hornsby, Sydney, NSW, Australia

20. Centre for Health Economics, Monash Business School, Monash University, Melbourne, Victoria, Australia

21. Eastern Health Clinical School, Monash University, Box Hill, Victoria, Australia

22. Orthopaedic Department, Royal Prince Alfred Hospital, Camperdown, Sydney, NSW, Australia

23. Bendigo Healthcare Group, Bendigo Hospital, Bendigo, Victoria, Australia

24. Orthopaedic Department, Fremantle Hospital, Fremantle, Perth, WA, Australia

25. Orthopaedic Department, Canterbury Hospital, Canterbury, Sydney, NSW, Australia

26. Orthopaedic Department, Calvary Hospital, Adelaide, SA, Australia

27. Orthopaedic Department, Fairfield Hospital, Fairfield, Sydney, NSW, Australia

28. Orthopaedic Department, St George Private Hospital, Kogarah, Sydney, NSW, Australia

29. Orthopaedic Department, Frankston Hospital, Frankston, Melbourne, Victoria, Australia

30. Orthopaedic Department, The Prince Charles Hospital, Chermside, Brisbane, QLD, Australia

31. Orthopaedic Department, Launceston General Hospital, Launceston, Tasmania, Australia

32. School of Medicine, St John of God Hospital and Barwon Health, Deakin University, Geelong, Australia

33. Orthopaedic Department, Mater Hospital, Raymond Terrace, Brisbane, QLD, Australia

34. Orthopaedic Department, Prince of Wales Hospital, Randwick, Sydney, NSW, Australia

35. Orthopaedic Department, Nepean Hospital, Nepean, Sydney, NSW, Australia

36. Orthopaedic Department, Coffs Harbour Base Hospital, Coffs Harbour, NSW, Australia

37. Orthopaedic Department, Western Health, Melbourne, Victoria, Australia

38. The Kolling Institute, Faculty of Medicine and Health, The University of Sydney and the Northern Sydney Local Health District, Sydney, NSW, Australia

39. Orthopaedic Department, Flinders Medical Centre, Bedford Park, Adelaide, SA, Australia

40. Department of Medicine and Public Health, Flinders University, Adelaide, SA, Australia

41. Orthopaedic Department, Sir Charles Gardiner Hospital, Perth, WA, Australia

42. Institute of Musculoskeletal Health, School of Public Health, The University of Sydney, Sydney, NSW, Australia

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
