## [Decision Letter · Decision Letter 0]

3 Dec 2023

PONE-D-23-32485Aspirin or Enoxaparin for VTE Prophylaxis after Primary Partial, Total or Revision Hip or Knee Arthroplasty: A Secondary Analysis from the CRISTAL Cluster Randomized TrialPLOS ONE

Dear Dr. Sidhu,

Thank you for submitting your manuscript to PLOS ONE. After careful consideration, we feel that it has merit but does not fully meet PLOS ONE’s publication criteria as it currently stands. Therefore, we invite you to submit a revised version of the manuscript that addresses the points raised during the review process.

We look forward to receiving your revised manuscript.

Kind regards,

Yoshihiro Fukumoto

Academic Editor

PLOS ONE

Journal Requirements:

"The trial was funded by an MRFF Grant provided by the Australian Federal Government (grant number 1152285)"          

"NO authors have competing interests"

Reviewers' comments:

Reviewer's Responses to Questions

**Comments to the Author**

1. Is the manuscript technically sound, and do the data support the conclusions?

Reviewer #1: Yes

Reviewer #2: Yes

Reviewer #3: Yes

2. Has the statistical analysis been performed appropriately and rigorously? 

Reviewer #1: Yes

Reviewer #2: Yes

Reviewer #3: Yes

3. Have the authors made all data underlying the findings in their manuscript fully available?

Reviewer #1: No

Reviewer #2: Yes

Reviewer #3: Yes

4. Is the manuscript presented in an intelligible fashion and written in standard English?

Reviewer #1: Yes

Reviewer #2: Yes

Reviewer #3: Yes

5. Review Comments to the Author

Reviewer #1: This is an interesting study to compare aspirin to enoxaparin for symptomatic VTE prophylaxis within 90 days of any type of hip or knee arthroplasty performed for any diagnosis, in patients enrolled in the CRISTAL trial. I have a few comments.

1. Is this analysis prespecified or post hoc? If prespecified, why did not the authors provide intonation in original JAMA paper? If this is a post hoc analysis, the authors need to state it. Furthermore, current format of manuscript looks like a first report of the CRRISTAL trial.

2. When looking into Table 1 and Figure 1, the numbers of patients are different from the original JAMA paper. The authors should explain how these numbers come.

Reviewer #2: First of all, the Reviewer would like to congratulate the authors for the current interesting study, which could give clinicians useful information for the issue. The primary publication was restricted to patients undergoing primary total hip or knee arthroplasty for a diagnosis of osteoarthritis, and provided a huge impact. This report included all enrolled patients undergoing hip or knee arthroplasty procedures for any indication. Although not as comprehensive as the main paper, this report also provides sufficient important information. The reviewer has no more comments for this paper.

Reviewer #3: This study presents a second analysis of the CRISTAL trial, examining the effectiveness of aspirin versus enoxaparin in preventing symptomatic venous thromboembolism (VTE) following hip or knee arthroplasty. The primary analysis of the CRISTAL trial, published in [Ref 3], focused on patients undergoing total hip or knee arthroplasty. In contrast to the primary CRISTAL publication, this study includes all types of hip or knee arthroplasty, aiming to improve the generalizability of the conclusions. Conducted across 31 Australian hospitals, the trial was a cluster-randomized crossover, registry-nested non-inferiority study. Patients received either 100mg of aspirin daily or 40mg of enoxaparin daily, with the treatment duration varying depending on the type of arthroplasty. The conclusion from this secondary analysis is that, for patients undergoing any type of hip or knee arthroplasty, enoxaparin is more effective than aspirin in reducing the risk of symptomatic VTE and joint-related reoperations within 90 days. The sample size and power of this study are well-justified. The statistical analysis is appropriate, and the conclusions are supported by the data.

One minor concern is that this study examined five different secondary outcomes and found a significant difference (p=0.03) in one secondary outcome. However, this may be due to multiple comparisons, and the result is likely to be insignificant after adjusting for multiple comparisons.

6. PLOS authors have the option to publish the peer review history of their article (what does this mean?). If published, this will include your full peer review and any attached files.

Reviewer #1: No

Reviewer #2: No

Reviewer #3: No

---

## [Author Response · Author response to Decision Letter 0]

1 Jan 2024

Please find attached authors' response document attached with submission.

Thank you

---

## [Decision Letter · Decision Letter 1]

22 Jan 2024

Aspirin or Enoxaparin for VTE Prophylaxis after Primary Partial, Total or Revision Hip or Knee Arthroplasty: A Secondary Analysis from the CRISTAL Cluster Randomized Trial

PONE-D-23-32485R1

Dear Dr. Sidhu,

We’re pleased to inform you that your manuscript has been judged scientifically suitable for publication and will be formally accepted for publication once it meets all outstanding technical requirements.

Kind regards,

Yoshihiro Fukumoto

Academic Editor

PLOS ONE

Additional Editor Comments (optional):

Reviewers' comments:

Reviewer's Responses to Questions

**Comments to the Author**

1. If the authors have adequately addressed your comments raised in a previous round of review and you feel that this manuscript is now acceptable for publication, you may indicate that here to bypass the “Comments to the Author” section, enter your conflict of interest statement in the “Confidential to Editor” section, and submit your "Accept" recommendation.

Reviewer #1: All comments have been addressed

Reviewer #3: All comments have been addressed

2. Is the manuscript technically sound, and do the data support the conclusions?

Reviewer #1: Yes

Reviewer #3: Yes

3. Has the statistical analysis been performed appropriately and rigorously? 

Reviewer #1: Yes

Reviewer #3: Yes

4. Have the authors made all data underlying the findings in their manuscript fully available?

Reviewer #1: Yes

Reviewer #3: Yes

5. Is the manuscript presented in an intelligible fashion and written in standard English?

Reviewer #1: Yes

Reviewer #3: Yes

6. Review Comments to the Author

Reviewer #1: (No Response)

Reviewer #3: (No Response)

7. PLOS authors have the option to publish the peer review history of their article (what does this mean?). If published, this will include your full peer review and any attached files.

Reviewer #1: No

Reviewer #3: No

---

## [Editor Report · Acceptance letter]

3 Apr 2024

PONE-D-23-32485R1 

PLOS ONE

Dear Dr. Sidhu, 

I'm pleased to inform you that your manuscript has been deemed suitable for publication in PLOS ONE. Congratulations! Your manuscript is now being handed over to our production team.

Kind regards, 

on behalf of

Dr. Yoshihiro Fukumoto 

Academic Editor

PLOS ONE